# Aircraft Maintenance Check Scheduling Using Reinforcement Learning

**Pedro Andrade** [1,*] **, Catarina Silva** [1] **, Bernardete Ribeiro** [1] **and Bruno F. Santos** [2]

1   Department of Informatics Engineering, University of Coimbra, CISUC, 3030-290 Coimbra, Portugal; catarina@dei.uc.pt (C.S.); bribeiro@dei.uc.pt (B.R.)
2   Air Transport and Operations, Faculty of Aerospace Engineering, Delft University of Technology, 2629 HS Delft, The Netherlands; b.f.santos@tudelft.nl
*   Correspondence: pmca@student.dei.uc.pt

**Abstract:** This paper presents a Reinforcement Learning (RL) approach to optimize the long-term scheduling of maintenance for an aircraft fleet. The problem considers fleet status, maintenance capacity, and other maintenance constraints to schedule hangar checks for a specified time horizon. The checks are scheduled within an interval, and the goal is to, schedule them as close as possible to their due date. In doing so, the number of checks is reduced, and the fleet availability increases. A Deep Q-learning algorithm is used to optimize the scheduling policy. The model is validated in a real scenario using maintenance data from 45 aircraft. The maintenance plan that is generated with our approach is compared with a previous study, which presented a Dynamic Programming (DP) based approach and airline estimations for the same period. The results show a reduction in the number of checks scheduled, which indicates the potential of RL in solving this problem. The adaptability of RL is also tested by introducing small disturbances in the initial conditions. After training the model with these simulated scenarios, the results show the robustness of the RL approach and its ability to generate efficient maintenance plans in only a few seconds.

**Keywords:** aircraft maintenance; maintenance check scheduling; reinforcement learning; q-learning





## 1. Introduction

Aircraft maintenance is the repair, modification, overhaul, inspection, and determination of the condition of aircraft systems, components, and structures that aims at providing aircraft airworthiness for operations [1].

Maintenance in aviation is usually defined using preventive strategies, which means that each aircraft must be grounded to perform a maintenance check at certain intervals. This strategy aims at constantly guaranteeing the airworthiness of the aircraft, reducing the risk of unexpected failures that are caused by a deterioration of components, and keeping a consistent maintenance program for the entire aircraft fleet. The maintenance check intervals that indicate when a new check is needed are dependent on the utilization as well as the aircraft aging. They are defined with respect to the aircraft usage parameters, which are defined in flight hours (FH), flight cycles (FC), and calendar days (DY). An FC corresponds to a complete take-off and landing sequence. Whenever an aircraft reaches the usage limit in one of the metrics, it must be grounded to perform the corresponding check. There are three major maintenance checks: A-checks, C-checks, and D-checks [2]. An A-check is a lighter check that is performed approximately every two to three months and it usually lasts one day. A C-check is a heavier check that is performed every 18 to 24 months and lasts between one and three weeks. A D-check, or structural check, is the heaviest check. It is performed every six to ten years and it lasts for three weeks to two months. In some literature, a B-check is also mentioned, although, in practice, most airlines opt to incorporate B-checks tasks into A-checks [3]. Because of the large impact these checks have on aircraft availability for flying and the amount of resources allocated to

perform these checks, they are typically planned months ahead for a time horizon of three to five years [3].

Maintenance costs represent around 10–20% of the total operating costs of an airline [4]. Despite this, the scheduling of aircraft maintenance is currently mostly based on a manual approach. Airlines rely on the experience of the maintenance personnel to plan the checks for fleets of aircraft, solving schedule conflicts that inevitably occur by shifting checks back and forth until a feasible calendar is found. Generally, this process is very time consuming, since it can take days or even weeks to develop a feasible maintenance plan. It also leads to sub-optimal solutions, which can affect aircraft availability, resulting in financial losses for the airline. Therefore, it is important to improve the current existing procedures and look towards maintenance optimization.

The literature on long-term aircraft maintenance scheduling is limited, as most of the research focus is on short-term aircraft maintenance routing. One of the main reasons is that the gains of optimizing the scheduling of checks can only be observed in the long term, and airlines usually aim for more rapid benefits. There are only two references in the literature addressing the long-term check scheduling problem. In [5], a simulation model was developed to reduce the maintenance costs and the time required to generate a maintenance plan. A second work is proposed in [3], where a Dynamic Programming (DP) based approach is used to schedule A and C-checks for a time horizon of four years while considering several maintenance constraints. The problem has a structure that follows the Markov Decision Process (MDP), and the goal is to schedule checks as close as possible to their due date, which is, the date when the respective interval is reached. By doing this, a higher FH utilization is achieved, which results in a lower amount of checks in the long run.

The aircraft maintenance routing problem consists in assigning aircraft to a sequence of flights that allow them to be routed to a maintenance station within a given number of days to perform a daily check [6]. One of the first studies on this topic is presented in [7], in which the goal is to build flight schedules that satisfy maintenance constraints, such as the maintenance base locations for the fleet. Over the years, several other approaches have been proposed. A polynomial time algorithm is developed in [8] to solve a maintenance routing problem, where each aircraft must visit a maintenance station every three days. In [9], the authors propose a solution for fleet allocation and maintenance scheduling using a DP approach and a heuristic technique. An Integer Linear Programming (ILP) model to generate feasible aircraft routes is developed in [10], to maximize fleet utilization. A set-partitioning problem was presented in [11] to minimize maintenance costs without violating the flight hour limit intervals of each aircraft. An approach for maximizing aircraft utilization by defining routes that satisfy the scheduled maintenance checks was proposed in [12]. A model to optimize the inventory management regarding the components and spare parts needed for maintenance was developed in [13]. The use of RL in maintenance scheduling problems was studied in [14]. The authors use the value function approximation and design policies for a dynamic maintenance scheduling problem, where the goal is to minimize the costs of expired maintenance tasks. In [15], RL algorithms were used to optimize the scheduling of maintenance for a fleet of fighter aircraft. The goal was to maximize availability and maintain it above a certain level. The results indicate that RL algorithms can find efficient maintenance policies, which have increased performance compared to heuristic policies.

In this paper, we develop a Deep Q-learning algorithm to solve the same check scheduling problem that is defined in [3]. By using RL, we can address some of the limitations found in the previous work. For instance, the time that is required to obtain a solution can be heavily reduced once the model is trained. RL can also answer some "curse of dimensionality" problems found with the DP-based method, and its adaptability is a relevant factor when dealing with this complex environment. At the same time, the check scheduling problem continues to be defined as an MDP. The model is trained using real maintenance data and, to evaluate the quality of the maintenance plans generated,

we perform a comparison with the work that was done in [3] and with airline estimations. One of the main advantages of RL is the possibility to adapt to changes over time without needing to restart the training process. This characteristic is verified by introducing disturbances in the initial conditions of the fleet and training the model with these simulated scenarios. The obtained results with this method indicate that RL can continue to produce good maintenance plans.

This paper is organized, as follows: Section 2 provides relevant background on RL, focusing on the Deep Q-learning algorithm used in this study. Section 3 presents the scheduling problem formulation. Section 4 describes the experimental setup. Section 5 depicts the key results that were obtained and their comparison with other approaches. Finally, Section 6 summarizes the research and addresses directions for future research on this topic.

## 2. Reinforcement Learning

In RL, an agent interacts with an environment by performing a series of subsequent decisions to reach a particular goal, receiving a reward depending on the quality of the action chosen [16]. The idea is that the agent will continuously learn by selecting actions that grant bigger rewards. The training of the RL agent occurs in several episodes. An episode is a sequence of states that starts on an initial state and ends on a terminal state.

Formally, RL can be described as a Markov Decision Process (MDP) and defined as a tuple (S, A, P, R, $\gamma$), where S is the set of states, A is the action set, P is the transition function, R is the reward function, and $\gamma$ is a discount factor, which is used to give less relevance to rewards occurring far into the future. At each time step $t$, the agent is in a state $s_t$ and it selects an action $a_t$. It then moves to the next state $s_{t+1}$ and receives a reward $r_t$, which indicates how good the selected action was. This interaction is represented in Figure 1. To select an action, the agent follows a policy, which is a mapping from states to actions. The policy defines the behavior of the agent.

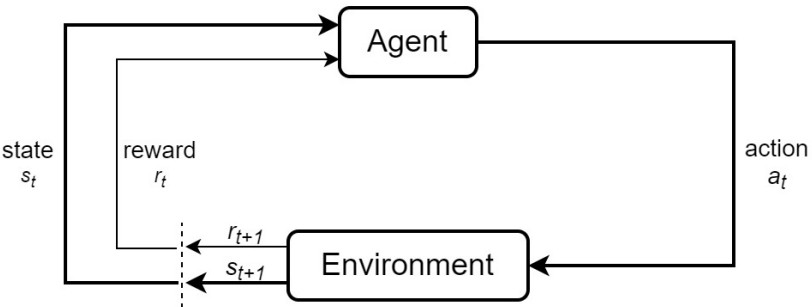

**Figure 1.** Interaction between the agent and the environment (adapted from [16]).

The goal is to find an optimal policy that corresponds to a sequence of actions that maximize the expected return, $R_t$. This return corresponds to the sum of discounted rewards over time and it can be defined with:

$$R_t = \sum_{k=0}^{\infty} \gamma^k r_{t+k}, \tag{1}$$

where $r_t$ is the reward obtained at time $t$ and $\gamma \in [0, 1]$ is the discount factor.

Another important element of RL is the value function, which defines the value of being in a given state. The value of a state $s$ when following a policy $\pi$, denoted $v_\pi(s)$, is the expected return when starting at state $s$ and following policy $\pi$ thereafter, and it can be formally defined as:

$$v_\pi(s) = \mathbb{E}_\pi[R_t \mid S_t = s] = \mathbb{E}_\pi \left[ \sum_{k=0}^{\infty} \gamma^k R_{t+k} \mid S_t = s \right] \tag{2}$$

There are two main methods for obtaining the optimal policy: policy iteration and value iteration. The first one aims to manipulate the policy directly, while the second one aims to find an optimal value function and adopt a greedy policy.

### 2.1. Q-Learning

Q-Learning [17] is based on value iteration and it is one of the most popular RL algorithms. It makes use of a look-up table, called Q-table, which, in most cases, has the shape [states, actions]. Each number in the Q-table represents the quality of taking an action, $a$, in a state, $s$, and it is named as the Q-value, $Q(s, a)$. The RL agent, at each time step, observes the current state and chooses the action with a higher Q-value in that state. After acting, the agent receives a reward, which will be used to update the Q-table using (3), where $\alpha$ is the learning rate.

$$Q(s_t, a_t) = Q(s_t, a_t) + \alpha \left[ r_{t+1} + \gamma \max_a Q(s_{t+1}, a_t) - Q(s_t, a_t) \right]. \tag{3}$$

The RL agent learns from his own decisions, which means that he must perform a large number of actions and collect a wide variety of experiences to be able to achieve a big reward and find the optimal policy. Suppose the agent adopts a greedy strategy by constantly choosing the action with the highest value. In that case, there is a risk of obtaining a suboptimal solution by converging to a local minimum. This is known as the exploration-exploitation trade-off, where the exploration corresponds to the agent choosing a random action, and the exploitation corresponds to the agent selecting the best action in the current state. The $\epsilon$-greedy is a widely known strategy for ensuring a proper exploration of the state-action space. The variable $\epsilon$ represents the probability of choosing a random action and it is usually initialized to 1 with a decay rate over time. This ensures ahigh exploration at the beginning, with exploitation rising over time.

While a look-up table can be a simple and efficient solution for state-action spaces of reduced size, the same is not true for bigger and more complex problems. The traditional solution in these cases is the Deep Q-learning variant, which uses Artificial Neural Networks (ANN) to obtain approximations for the Q-values. Experience replay is a relevant feature frequently used with Deep Q-learning. Instead of training the network with the sequence of experiences as they occur during the simulation, those experiences are saved in what is usually called the replay memory. After each action of the agent, a random batch of experiences is sampled from the replay memory and used for training [18]. This helps to overcome two common problems of Deep Q-learning: the agent forgetting past experiences as time passes, and the correlation between consecutive experiences.

### 3. Problem Formulation

In the aviation industry, each aircraft must be grounded to perform maintenance checks at certain intervals, as specified in the aircraft Maintenance Planning Document (MPD). These intervals are defined in flight hours (FH), flight cycles (FC), and calendar days (DY), which are the three metrics that represent the aging of the aircraft. Usually, each check type has its own maximum defined interval, and the corresponding usage metrics are reset to 0 immediately after a check is performed.

It is possible to schedule maintenance checks beyond the maximum interval by using a tolerance, which is also defined in the aircraft MPD. However, the use of this tolerance should be avoided since permission has to be requested from the civil aviation authorities. Furthermore, the FH, FC, and DY that are used from the tolerance have to be deducted from the maximum interval metrics of the next check.

In cases where a lighter check is scheduled near a heavier one, airlines frequently opt to merge the two of them. For instance, it is common for an A-check to be merged into a C-check if both of them reach their interval within a certain number of days from each other. This merging prevents the same aircraft from being grounded twice in a short period

without necessarily increasing the duration of the C-check. The disadvantage of doing this is the increase in the number of A-checks in the long term.

Some airlines also require a minimum number of days between the start dates of two checks of the same type. This is mainly related to resource preparation, such as workforce, tools, aircraft spare parts, or the hangar itself.

### 3.1. Assumptions

We consider the following assumptions to define our long-term maintenance check scheduling approach:

**A1**     The minimum time unit for the scheduling of maintenance is 1 DY.
**A2**     Aircraft utilization can be measured in FH, FC, and DY.
**A3**     The daily utilization of each aircraft can be defined by the airline or can be estimated using historical data.
**A4**     The duration of each check can be defined by the airline or can be estimated using historical data.
**A5**     There is a limited amount of hangar slots each day to perform A/C-checks.
**A6**     Each A/C-check uses only one hangar slot for its total duration.
**A7**     C-checks have a higher priority than A-checks.
**A8**     An A-check can be merged in a C-check without increasing the C-check duration and without occupying an A-check slot.
**A9**     The fleet operation plan does not influence the scheduling of maintenance.

In addition to **A5**, it is relevant to know that the available maintenance slots can vary during the year. For instance, during commercial peak seasons, the amount of maintenance performed is greatly reduced, and heavy maintenance can even be suspended. These peak seasons usually consist of Christmas, New Year, Easter, and Summer periods.

The priority that is given to the scheduling of C-checks mentioned in **A7** has two main reasons. The first one is related to the possibility of merging checks. When planning the A-checks, it is important to know the exact starting and ending days of the C-checks to decide whether the merging is possible. The second reason is related to the fact that the A-check usage parameters are not increased when the aircraft is performing a C-check (due to it being grounded). Because a C-check usually involves several weeks of ground time, it has great influence on the scheduling of the A-checks.

There is a deviation from traditional state-of-the-art problems by not considering the operation plan, as mentioned in **A9**. The reason is that maintenance checks have intervals of several months/years, while the airline only plans aircraft routes for the next few days. Consequently, the geographic location of the aircraft is also not considered.

### 3.2. Decision Variable

The decision variable in this problem corresponds to selecting an aircraft to have its next check scheduled (giving priority to C-checks). At each step $j$, the decision variable $x_j$ can be defined with (4), where $i$ is the chosen aircraft index.

$$x_j = i, \quad i \in \{1, ..., N\}, \ j \in \{0, ..., J\} \tag{4}$$

### 3.3. State Variables

The state vector $s_j$ includes a set of attributes that influence the decisions:

$$s_j = \{m_{h,t}^k, \ M_t^k, \ DY_{i,t}^k, \ FH_{i,t}^k, \ FC_{i,t}^k, \ \sigma_{i,k}^{DY}, \ \sigma_{i,k}^{FH}, \ \sigma_{i,k}^{FC}, \ D_i^k, \ u_i^k, \ y_i^k, \ z_i^k, \ g_{i,t}^k, \ G_{i,t}^k, \ L_i^k, \ \delta_i\} \tag{5}$$

where each attribute contains information of aircraft $i \in \{1, ..., N\}$, on a specific day $t \in \{0, ..., T\}$, with respect to a check type $k \in \{A, C\}$. These attributes are defined for each aircraft for the entire time horizon. It is important to note that only working days are considered, that is, days in which hangar maintenance can be performed.

- $m_{h,t}^k$ is a binary variable to indicate if a type $k$ check can be performed in hangar $h$ on day $t$.
- $M_t^k$ is the hangar capacity for type $k$ checks on day $t$.
- $DY_{i,t}^k$, $FH_{i,t}^k$, and $FC_{i,t}^k$ correspond to the cumulative usages in DY, FH, and FC, respectively, of aircraft $i$ on day $t$ for a type $k$ check.
- $\sigma_{i,k}^{DY}$, $\sigma_{i,k}^{FH}$, and $\sigma_{i,k}^{FC}$ correspond to the tolerance used in the last type $k$ check of aircraft $i$, with respect to the same three metrics.
- $D_i^k$ is the next due date for type $k$ check of aircraft $i$, that is, the day in which the respective interval is reached.
- $u_i^k$ is the next due date for type $k$ check of aircraft $i$ using the tolerance.
- $y_i^k$ is the ending day of the last type $k$ check for aircraft $i$.
- $z_i^k$ is the scheduled day of the next type $k$ check for aircraft $i$
- $g_{i,t}^k$ is a binary variable to indicate if aircraft $i$ is grounded on day $t$ performing a type $k$ check.
- $G_{i,t}^k$ is a binary variable to indicate if there is a type $k$ check starting on day $t$ for aircraft $i$.
- $L_i^k$ is the duration (in working days) of the next type $k$ check for aircraft $i$.
- $\delta_i$ is the amount of FH lost in the last scheduled check of aircraft $i$.

There are also fixed attributes, such as check intervals and tolerances, aircraft average utilization, and others that are predefined by the airline, such as the minimum number of days between the start of two type $k$ checks, and other constraint variables. All of these attributes are described in the Nomenclature section at the end of the document.

### 3.4. State Transition

The state transition defines a set of equations that determine how the state attributes evolve along each step. With every new action, an aircraft is selected and its next A/C-check is scheduled. Several conditions must be verified to choose the day in which to schedule the check. The first one is that there needs to be hangar availability during the entire duration of the check. We define the capacity for a particular hangar $h$ as:

$$m_{h,t}^k = \begin{cases} 1, & \text{if hangar } h \text{ is available for type } k \text{ checks on day } t \\ 0, & \text{otherwise} \end{cases} \tag{6}$$

This variable is used to calculate whether there is enough hangar availability for type $k$ checks on a specific day $t$:

$$M_t^k = \begin{cases} 1, & \text{if } \sum_h m_{h,t}^k > 0 \\ 0, & \text{otherwise} \end{cases} \tag{7}$$

Finally, the hangar capacity for the entire duration of the check can be defined with:

$$H_{i,t}^k = \begin{cases} 1, & \text{if } L_i^k - \sum_{t'=t}^{t+L_i^k} M_{t'}^k = 0 \\ 0, & \text{otherwise} \end{cases} \tag{8}$$

where $t$ is the candidate day to schedule the check.

A second condition is related to the minimum number of days between the start dates of two consecutive type $k$ checks, $w_k$. We can use $G_{i,t}^k$ to calculate the total number of checks

starting in the range $[t - w_k, t + w_k]$, with $t$ being the candidate day to schedule the check. If this sum is equal to 0, then a type $k$ check for aircraft $i$ can be scheduled on day $t$:

$$G_t^k = \begin{cases} 1, & \text{if } \sum_i \left( \sum_{t'=t-w_k}^{t+w_k} G_{i,t'}^k \right) = 0 \\ 0, & \text{otherwise} \end{cases} \tag{9}$$

With Equations (8) and (9), it is possible to select the day in which to schedule the next $k$ check of aircraft $i$, defined as $z_i^k$:

$$z_i^k = \max\left\{ \left( t_{min} G_{t_{min}}^k H_{i,t_{min}}^k \right), \ \left( (t_{min}+1) G_{t_{min}+1}^k H_{i,t_{min}+1}^k \right), \ \ldots, \ \left( t_{max} G_{t_{max}}^k H_{i,t_{max}}^k \right) \right\} \tag{10}$$

where $t_{min} = y_i^k$ and $t_{max} = D_i^k$. By default, the use of tolerance should always be avoided in the planning phase. Therefore, the goal with Equation (10) is to choose the maximum day possible, which is, the closest possible day to the check due date, so that we minimize the unused FH of the aircraft. If the value of $z_i^k$ obtained is equal to 0, then there are no possible days to schedule the check before the due date. This can happen when there is not enough hangar availability and/or the spacing rule between checks is not verified. In this case, the tolerance can be used and the check scheduled day is defined with (11):

$$z_i^k = \max\Big\{ (t_{max}+1-t_{min}) G_{t_{min}}^k H_{i,t_{min}}^k,$$
$$(t_{max}+1-t_{min}+1) G_{t_{min}+1}^k H_{i,t_{min}+1}^k,$$
$$\ldots,$$
$$(t_{max}+1-t_{max}) G_{t_{max}}^k H_{i,t_{max}}^k \Big\} \tag{11}$$

where $t_{min} = D_i^k$ and $t_{max} = u_i^k$. Similarly, the goal is to choose a day as close as possible to the check due date, but this time, with the goal of reducing the amount of tolerance used.

Several attributes need to be updated after calculating the scheduled day for the next check of aircraft $i$. The hangar capacity during the check is reduced since a new aircraft is now occupying a slot:

$$\tilde{m}_{h,t}^k = \begin{cases} m_{h,t}^k - 1, & \text{if } t \in [z_i^k, z_i^k + L_i^k] \\ m_{h,t}^k, & \text{otherwise} \end{cases} \tag{12}$$

Attributes $G_{i,t}^k$, which indicates the check start days for aircraft $i$, and $g_{i,t}^k$, which indicates the days in which aircraft $i$ is grounded, are updated with:

$$\tilde{G}_{i,t}^k = \begin{cases} 1, & \text{if } t = z_i^k \\ G_{i,t}^k, & \text{otherwise} \end{cases} \tag{13}$$

$$\tilde{g}_{i,t}^k = \begin{cases} 1, & \text{if } t \in [z_i^k, z_i^k + L_i^k] \\ g_{i,t}^k, & \text{otherwise} \end{cases} \tag{14}$$

Now that a new type $k$ check is scheduled for aircraft $i$, we can set $y_i^k$ to the last day of this recently scheduled check:

$$y_i^k = z_i^k + L_i^k \tag{15}$$

The amount of FH that is lost as a result of this scheduling, defined as $\delta_i$, can be computed with Equation (16).

$$\delta_i = \left| I_{i,k}^{FH} - FH_{i,t}^k \right| \tag{16}$$

Subsequently, the cumulative utilization attributes of aircraft $i$, $DY_{i,t}^k$, $FH_{i,t}^k$, and $FC_{i,t}^k$, are set to 0 on day $y_i^k + 1$. The duration of the next check, $L_i^k$, is updated according to a lookup table that contains the duration of future checks specified by the airline.

The last step in the state transition is to calculate the new due date for a type $k$ check. We start by updating the amount of tolerance used in the last type $k$ check, which must be deducted from the next interval:

$$\sigma_{i,k}^{DY} = \max\left\{0, \, DY_{i,t}^k - (D_i^k - y_i^k)\right\} \tag{17}$$

$$\sigma_{i,k}^{FH} = \max\left\{0, \, FH_{i,t}^k - (D_i^k - y_i^k) \times fh_i\right\} \tag{18}$$

$$\sigma_{i,k}^{FC} = \max\left\{0, \, FC_{i,t}^k - (D_i^k - y_i^k) \times fc_i\right\} \tag{19}$$

where $fh_i$ and $fc_i$ correspond to the average daily utilization of aircraft $i$ estimated by the airline in FH and FC, respectively, and $t = z_i^k$. We also need to consider the number of days in which the aircraft is grounded and its utilization does not increase. For instance, the grounded days to perform the C-check do not count to compute the next A-check due date if a C-check is scheduled after the last A-check. We define the day in which the interval for the next type $k$ check is reached as $d_{i,k}^{DY}$, $d_{i,k}^{FH}$, and $d_{i,k}^{FC}$, with respect to each usage metric, without yet considering the aircraft ground time:

$$d_{i,k}^{DY} = y_i^k + I_{i,k}^{DY} - \sigma_{i,k}^{DY} \tag{20}$$

$$d_{i,k}^{FH} = y_i^k + \frac{I_{i,k}^{FH} - \sigma_{i,k}^{FH}}{fh_i} \tag{21}$$

$$d_{i,k}^{FC} = y_i^k + \frac{I_{i,k}^{FC} - \sigma_{i,k}^{FC}}{fc_i} \tag{22}$$

where $I_{i,k}^{DY}$, $I_{i,k}^{FH}$, and $I_{i,k}^{FC}$ are the type $k$ check intervals in DY, FH, and FC, respectively. Subsequently, the due date of the next type $k$ check is defined with:

$$D_i^k = \min\left\{ d_{i,k}^{DY} + \sum_{t=y_i^k}^{d_{i,k}^{DY}}\sum_k g_{i,t}^k, \, d_{i,k}^{FH} + \sum_{t=y_i^k}^{d_{i,k}^{FH}}\sum_k g_{i,t}^k, \, d_{i,k}^{FC} + \sum_{t=y_i^k}^{d_{i,k}^{FC}}\sum_k g_{i,t}^k \right\} \tag{23}$$

This due date corresponds to the day in which the first usage interval is reached with the addition of the aircraft ground time since the last type $k$ check.

The last attribute to update is the due date for the next $k$ check when the tolerance is used, $u_i^k$:

$$u_i^k = \min\left\{ D_i^k + \theta_{i,k}^{DY}, \, D_i^k + \frac{\theta_{i,k}^{FH}}{fh_i}, \, D_i^k + \frac{\theta_{i,k}^{FC}}{fc_i} \right\} \tag{24}$$

where $\theta_{i,k}^{DY}$, $\theta_{i,k}^{FH}$, and $\theta_{i,k}^{FC}$ are the maximum tolerances that are allowed for a type $k$ check, with respect to DY, FH, and FC, respectively.

### 3.5. Problem Constraints

There are multiple constraints in this check scheduling problem and some of them were already defined in the previous subsections. One of the most important conditions in this problem is to guarantee that the aircraft usage conditions relative to each check type never exceed the maximum allowed. This maximum corresponds to the sum of the check interval with the fixed tolerance while deducting the amount of tolerance that is used in the last check. This constraint can be formulated for each usage metric with the following three equations:

$$DY_{i,t}^k \leq I_{i,k}^{DY} + \theta_{i,k}^{DY} - \sigma_{i,k}^{DY} \tag{25}$$

$$FH_{i,t}^k \leq I_{i,k}^{FH} + \theta_{i,k}^{FH} - \sigma_{i,k}^{FH} \tag{26}$$

$$FC_{i,t}^k \leq I_{i,k}^{FC} + \theta_{i,k}^{FC} - \sigma_{i,k}^{FC} \tag{27}$$

It is also necessary to guarantee that the number of checks occurring in parallel in each day does not exceed the available number of hangar slots:

$$\sum_t \sum_i g_{i,t}^k \leq \sum_h m_{h,t}^k, \quad t \in \{t_0, ..., T\}, \quad i \in \{1, ..., N\}, \tag{28}$$

where $g_{i,t}^k$ is a binary variable to indicate if aircraft $i$ is grounded on day $t$ performing a type $k$ check, and $m_{h,t}^k$ is a binary variable to indicate if a type $k$ check can be performed in hangar $h$ on day $t$.

Finally, we formulate the constraint relative to the minimum number of days between the start of two consecutive type $k$ checks:

$$\sum_i \sum_{t'=t-w_k}^{t+w_k} G_{i,t'}^k \leq 1, \quad t \in \{t_0, ..., T\}, \quad i \in \{1, ..., N\}, \tag{29}$$

where $G_{i,t'}^k$ is a binary variable to indicate whether there is a type $k$ check starting on day $t'$ for aircraft $i$, and $w_k$ is the minimum number of DY between the start of two type $k$ checks.

### 3.6. Objective Function

Cost reduction is one of the main goals in aviation maintenance. In a long term maintenance scheduling problem, the best way to meet this goal is to minimize the unused FH of the fleet. This results in a reduction of the number of checks scheduled in the long run, which can have a significant impact since each day out of operations has a great cost for the airline. At each step of the scheduling algorithm, we can calculate the cost of choosing action $x_j$ on state $s_j$, $C_j(s_j, x_j)$, with:

$$C_j(s_j, x_j) = \begin{cases} \delta_i, & \text{if } \delta_i \geq 0 \\ -\delta_i\, P, & \text{if } \delta_i < 0 \end{cases} \tag{30}$$

where $i$ is the selected aircraft, $\delta_i$ is the unused FH in the last scheduled check of aircraft $i$, and $P$ is a penalty for using the tolerance.

The objective function corresponds to minimizing this cost and can be defined with Equation (31), where $X^\pi(s_j)$ is the optimal scheduling policy function. This optimal scheduling policy consists of selecting the best action in any given state, which is, the action that yields a lower cost.

$$\min_\pi \mathbb{E}\left\{ \sum_{j=0}^J C_j\big(s_j, X^\pi(s_j)\big) \right\} \tag{31}$$

## 4. Experimental Setup

This section details the developed Deep Q-learning algorithm, with a special focus on the RL elements and the algorithm parameters. It also presents the real maintenance dataset and the two test cases that are used to validate the solution, representing real and simulated scenarios.

### 4.1. Deep Q-Learning

In this work, we propose a Deep Q-learning algorithm to optimize a long-term check scheduling problem. Figure 2 presents the general workflow of the scheduling process. At each decision step, the RL agent chooses an aircraft $i$, and the next type $k$ check for that aircraft is scheduled to the closest available day to its due date. As mentioned before,

C-checks are scheduled first to deal with merging opportunities. Subsequently, the due date for the next type *k* check is calculated, and the necessary aircraft information is updated.

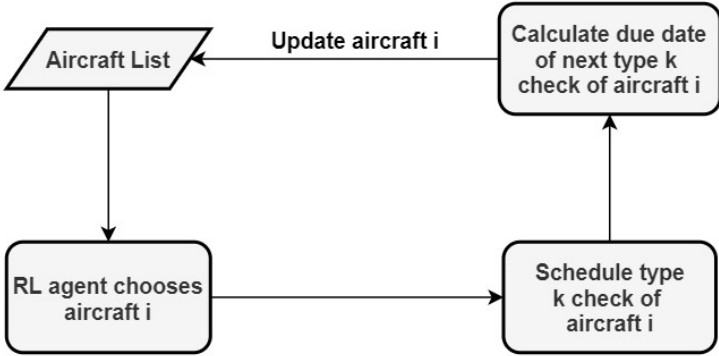

**Figure 2.** General workflow of the check scheduling process.

The agent is trained using experience replay, and the amount of exploration is controlled using an $\epsilon$-greedy strategy. In the first episode, $\epsilon$ is equal to 1, meaning that the agent will always prioritize exploration. This value has a linear decay rate over each episode until it reaches 0.01.

The Double Q-learning method that was introduced in [19] is also used. This method applies a double estimator approach to determine the value of the next state. The main benefit is the removal of over-estimations that are known to occur in the action values. We follow the approach that was proposed in [20] by adding a second network, called the target network. It is used to estimate the action values, while the online network is used to select the action greedily. Periodically, the weights of the target network are updated by being replaced by the weights from the online network.

Algorithm 1 presents the pseudo-code for the Deep Q-learning algorithm.

---

**Algorithm 1:** Pseudo-code for the Deep Q-Learning algorithm.

---

Initialize replay memory D;
Initialize the online network $Q^A$ with random weights;
Initialize the target network $Q^B$ with random weights;
**for** *episode* $= 1, M$ **do**
    $s_j = s_1$;
    **for** $j = 1, J$ **do**
        $a_j$ = random action with probability $\epsilon$, otherwise,
        $a_j = argmax_a Q^A(s_j, a_j)$;
        Simulate action $a_j$ and observe the reward $r_j$ and the next state $s_{j+1}$;
        Store the tuple $(s_j, a_j, r_j, s_{j+1})$ in D;
        $s_j = s_{j+1}$;
        Sample a minibatch of $(s_{j'}, a_{j'}, r_{j'}, s_{j'+1})$ from D to train $Q^A$;
        Compute the target value, $y_{j'} =$
$$\begin{cases} r_{j'}, \text{if the episode terminates at step j'+1} \\ r_{j'} + \gamma max_a Q^B(s_{j'+1}, a), \text{otherwise} \end{cases};$$
        Calculate the loss using $(y_{j'} - Q^A(s_{j'}, a_{j'}))^2$ and update the weights of $Q^A$;
        **if** $Update(Q^B)$ **then**
            Replace $Q^B$ weights with $Q^A$ weights;
        **end**
    **end**
**end**

---

First, an empty queue is created to represent the replay memory. The agent experiences are stored in the replay memory and used to train the model, as explained in Section 2.1. The online and target networks are initialized using Xavier initialization [21]. At the beginning of each episode, the current state is set to the initial state. Subsequently, for each step, an action is chosen either randomly or greedily with a certain probability. Each action corresponds to choosing a specific aircraft, as mentioned previously. Subsequently, the action is simulated, meaning that the next check for the respective aircraft is scheduled. The reward and next state are calculated and stored in the replay memory, and the current state advances to the next state. A minibatch is sampled from the replay memory and used to train the online network by computing the target value and respective loss. This loss is the minimum squared error of the predicted value and the target value. Finally, the target network can be updated by replacing its weights with the weights from the online network.

The reward that the agent obtains depends on the amount of FH lost with the scheduling of a check. The reward function can be defined with Equation (32):

$$R = \begin{cases} fh_i\,(z - D), & \text{if } 0 \leq z \leq D \\ 10\,fh_i\,(D - z), & \text{if } 0 \leq D < z \\ -10^5, & \text{if } z = -1 \end{cases} \tag{32}$$

where $z$ is the start day of the check, $D$ is its due date, and $fh_i$ is the average daily utilization of aircraft $i$ in FH. The function contains three conditions that represent three penalty levels for the agent. The first condition corresponds to the smallest penalty, which is equal to the FH lost when the check is scheduled without using the tolerance. The second one is equal to the amount of tolerance FH that is used multiplied by a factor of 10, to punish the agent more for using the tolerance. The final condition is the highest penalty and it represents a scenario in which the check was not scheduled.

The neural network and the hyper-parameters for the Deep Q-learning algorithm were obtained using a grid search technique. For each parameter, several values were defined and individually tested to see the ones that produced better results. The chosen network consists of a multilayer perceptron with two fully connected hidden layers, the first with 400 neurons and the second with 100 neurons. Both of the layers have the sigmoid activation function and the optimizer used is the Adam optimizer [22]. Table 1 presents the remaining hyper-parameters.

**Table 1.** Hyper-parameters for the Deep Q-learning algorithm.

| Hyper-Parameter | Value | Description |
|---|---|---|
| Episodes | 200 | Total number of training episodes |
| Max steps | 10,000 | Maximum number of agent steps per episode |
| Replay memory size | 100,000 | Size of the queue containing the agent experience |
| Batch size | 32 | Number of samples taken from replay memory |
| Discount factor | 0.99 | Discount factor of future rewards |
| Learning rate | 0.0001 | Learning rate used by the optimizer |
| Initial $\epsilon$ | 1 | Initial value for exploration |
| Final $\epsilon$ | 0.01 | Final value for exploration |
| Target update | 10,000 | Step frequency to update the target network |

*4.2. Test Cases*

Two test cases are defined based on real and simulated scenarios to evaluate the proposed approach.

### 4.2.1. Real Scenario

The first test case corresponds to a real maintenance scenario that consists of 45 aircraft from the Airbus A320 family (A319, A320, and A321). All aircraft share the same A/C-check interval and tolerances, which are presented in Table 2.

**Table 2.** A/C-check interval and tolerance for the Airbus A319, A320, and A321.

|  | **Calendar Days** | **Flight Hours** | **Flight Cycles** |
|---|---|---|---|
| A-check interval | 120 | 750 | 750 |
| A-check tolerance | 12 | 75 | 75 |
| C-check interval | 730 | 7500 | 5000 |
| C-check tolerance | 60 | 500 | 250 |

The duration of the A-checks is always one calendar day, and the airline provides the duration of the next five C-checks for each aircraft. The minimum number of days between the start dates of two consecutive checks of the same type, defined previously as $w_k$, is equal to three for the C-checks and 0 for the A-checks. The dataset also contains aircraft usage estimations. On average, an aircraft has a daily utilization of 10.5 FH and 4.7 FC. The maintenance slots for A/C-checks are also fixed. For the A-checks, there is one slot on Mondays and Thursdays and two slots on Tuesdays and Wednesdays. For the C-checks, there are three slots every working day. However, the C-check work is interrupted during commercial peak seasons. These consist of the summer period between the 1st of June and the 30th of September, and the Christmas, New Year, and Easter periods. There is also information about the initial condition of the fleet, namely the usage of each aircraft since their last A and C-check.

The goal in this scenario is to find an optimal solution for long-term A/C-check scheduling. The horizon goes from 25 September of 2017 to 31 December of 2021 [23]. To evaluate the quality of the RL solution, we compare it to the work that was presented in [3], which proposed a DP based approach for solving the same scheduling problem using the same dataset.

### 4.2.2. Simulated Scenarios

The second test case aims to evaluate the RL solution's ability in dealing with different initial conditions of the fleet. We use the same dataset as a baseline for the same time horizon, and introduce small disturbances in the initial usage metrics of the fleet, which is, the aircraft usage in DY, FH, and FC since its last A and C-check. Although, it is necessary to limit how much the aircraft usage can be modified. The airline seeks a balanced maintenance demand for the fleet. Checks are usually planned to avoid any future problems with multiple aircraft competing for limited maintenance slots. Therefore, if we drastically change the initial fleet conditions, it is likely that the scheduling is not possible under the specified maintenance constraints.

The simulation of new initial usages for the fleet is done at three levels. The first level defines how much the initial usage will change. For each aircraft, a random value between 1% and 10% is chosen, which means that the aircraft usage (in DY, FH, and FC) will increase or decrease by that amount. The reason for the upper bound of 10% is that, when testing with higher values, the scheduling would often be unfeasible. The second level defines whether the amount of usage percentage chosen is added or subtracted to the actual aircraft usage. Finally, in the third level, the initial fleet usages are shuffled. The idea is to allow the agent to learn which action to take based on the aircraft conditions rather than their id.

We simulated 100 new initial conditions for the fleet, which were used to train and test the model. The training set consisted of 90 simulated scenarios and the testing set contained the remaining 10. The training of each simulated scenario is done in 200 episodes using the same parameters mentioned previously.

## 5. Results

This section presents the results that were obtained in both test cases. A set of Key Performance Indicators (KPI) is defined to evaluate the quality of the maintenance plan generated and it consists of:

- the number of A and C-checks scheduled during the entire horizon,
- the average FH used between two consecutive A and C-checks,
- the number of A and C-checks scheduled using the tolerance, and
- the number of A-checks merged into C-checks.

These KPIs are all relevant in evaluating the maintenance plan, since they can all influence maintenance costs, which airlines aim to minimize. Regarding the RL algorithm, one of the most important elements to evaluate the performance of the agent is the learning process. Therefore, the sum of rewards obtained by the agent at the end of each training episode is a good performance indicator.

### 5.1. Real Scenario

In order to evaluate the quality of the maintenance plan generated for the real scenario, a comparison to the work done in [3] is performed, in which the authors developed a practical DP methodology to solve the same problem. The results are also compared with airline estimations for the period. However, the A-check metrics were not available for the airline, because they only plan A-checks for the next year. Table 3 presents the comparison of the three approaches.

**Table 3.** Comparison of the A and C-check scheduling results between the three approaches under several Key Performance Indicators.

| KPI | Airline | DP | RL |
|---|---|---|---|
| Total C-checks | 96 | 88 | 86 |
| C-check Average FH | <6600 | 6615 | 7122 |
| Tolerance Events | 6 | 4 | 1 |
| Total A-checks | 895–920 | 877 | 876 |
| A-check Average FH | — | 717.6 | 716.8 |
| Tolerance Events | — | 0 | 0 |
| Merged A-checks | — | 18 | 14 |
| Computation Time (s) | ≥3 days | 510 | 781 |

The results indicate an improvement of both the DP and the RL approaches over the airline in all of the KPI considered, meaning that both methods can generate a better maintenance plan much faster.

For the C-checks, the results show that the RL solution produces better results. The higher C-check average FH means that the average usage of the fleet in between two consecutive C-checks is higher, leading to two fewer checks scheduled in the time horizon. This outcome has great significance if we think that each C-check represents several days of ground time for an aircraft, without generating any profit. Regarding the A-checks, the results of the RL and DP solutions are very similar in both the total number of A-checks scheduled and in the average usage of the fleet.

Figures 3 and 4 show the distribution of A and C-check usages for the entire fleet. As we can see, in the optimal maintenance plan generated by the RL solution, the majority of A and C-checks are scheduled close to their interval.

There are a few checks with lower utilization due to limitations in hangar maintenance at some periods. For instance, there are two C-checks with less than 5000 FH, which had their due date during the Christmas period. The C-check constraints during this period and the Summer season lead the algorithm to schedule them several months before their due date.

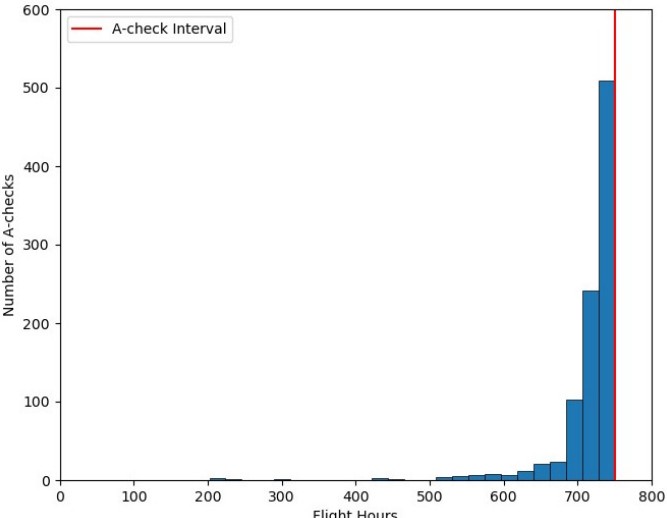

**Figure 3.** Distribution of A-check usages.

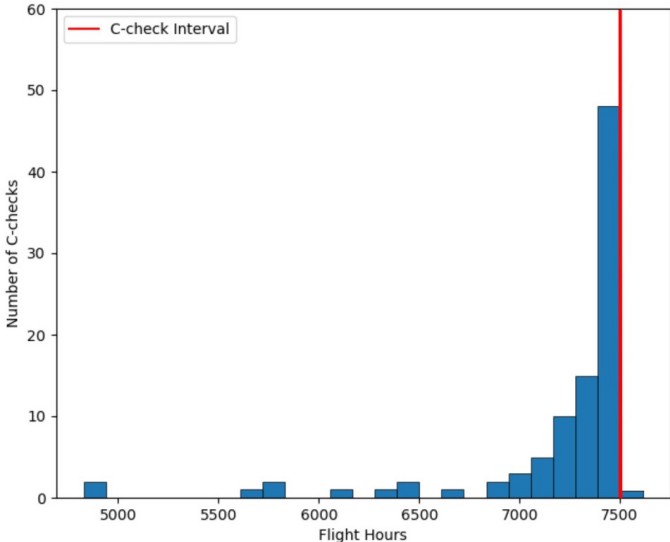

**Figure 4.** Distribution of C-check usages.

Figure 5 presents the sum of rewards that were obtained by the agent over each training episode. In the beginning, the lack of experiences collected by the agent, combined with higher exploration rates, results in lower rewards. Although, as the agent learns, the obtained rewards get higher over time, until it reaches convergence around episode 140.

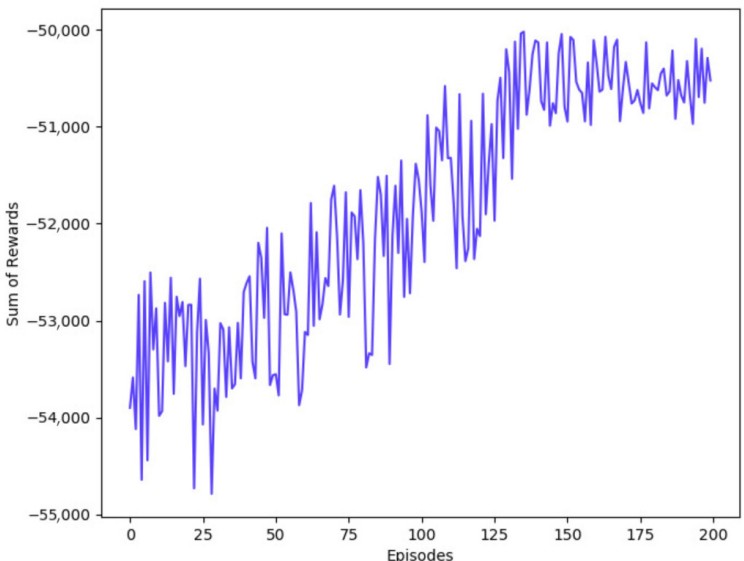

**Figure 5.** Sum of agent rewards during training.

*5.2. Simulated Scenarios*

In this stage, small disturbances were introduced in the initial fleet conditions to generate different maintenance scenarios. In particular, aircraft usages since their last A and C-check were randomized according to the three previously explained levels. The same RL algorithm is used to train a model with these simulated scenarios. The idea is to evaluate the ability of the RL solution in generating good maintenance plans when the fleet conditions change. Table 4 presents the mean and standard deviation that were obtained for the set of scenarios used for testing, regarding the established KPIs.

**Table 4.** The results obtained with simulated scenarios.

| KPI | Real Scenario | Simulated Scenarios | |
|---|---|---|---|
| | RL | Mean | Std. Dev |
| Total C-checks | 86 | 90.38 | 2.21 |
| C-check Average FH | 7122 | 6958.17 | 84.95 |
| Tolerance Events | 1 | 1.28 | 1.96 |
| Total A-checks | 876 | 892.24 | 15.47 |
| A-check Average FH | 716.8 | 713.83 | 3.61 |
| Tolerance Events | 0 | 0.71 | 1.01 |
| Merged A-checks | 14 | 15.37 | 2.23 |

If we analyze these results with the RL results for the real scenario, we see a decrease in the quality of the solution. This is more noticeable when scheduling C-checks due to the higher number of constraints. The decrease in quality is to be expected since the model is not specialized in a single scenario. Although, the average FH of the fleet for both A and C-checks remains close to the respective intervals. The low amount of tolerance events means that the use of tolerance when scheduling the checks is being avoided. Finally, the number of merged checks also remains consistent. Overall, it is safe to say that the RL solution can produce quality maintenance plans for different fleet scenarios.

An advantage of RL over other traditional approaches in solving this type of problem is the computation time for generating a solution after the model is trained. In this case, the training of the model lasted close to 20 h. After that, a new solution could be only be obtained in a few seconds. A fast planning tool can add significant value in such a high uncertainty environment in dealing with operation disruptions or other unexpected events. Furthermore, the agent can continue to learn and improve the maintenance plans that are produced over time.

## 6. Conclusions

This paper proposed a reinforcement learning solution to optimize the long-term scheduling of A and C-checks for an aircraft fleet. The goal was to maximize the FH usage between checks, which allows for an increase in aircraft availability in the long run. The proposed approach uses a Deep Q-learning algorithm with experience replay and the Double Q-learning variant. At each decision step, the RL agent selects an aircraft to have its next check scheduled on the best available day, prioritizing the scheduling of C-checks.

This approach is evaluated in a real scenario using data from a fleet of 45 aircraft from three sub-fleets (A319, A320, and A321). A comparison with airline estimations for the same period shows that the RL solution produces more efficient maintenance plans. The higher fleet usage for both A and C-checks results in a lower amount of checks being scheduled during the entire time horizon, which has significant financial benefits for the airline. The RL results are also promising when compared to the DP approach, achieving better results in the C-check scheduling.

A second test case is defined to evaluate the ability of RL in dealing with different initial conditions of the fleet. Multiple scenarios are simulated by introducing disturbances in the initial usage metrics of each aircraft. These scenarios are used to train the RL agent with different fleet conditions. The results obtained with several testing scenarios show that this approach can generate quality maintenance plans in a short amount of time.

There are not many studies regarding long-term maintenance problems in aviation despite its great relevance, which encourages future research on this topic. A potential opportunity is to extend the check scheduling algorithm by defining the set of maintenance tasks to be performed in each one of them. This task allocation problem would produce an optimal maintenance plan with a combination of maintenance checks and tasks.

**Author Contributions:** Conceptualization, P.A., C.S., B.R., and B.F.S.; software, P.A.; writing—original draft preparation, P.A.; writing—review and editing, P.A., C.S., B.R., and B.F.S.; supervision, C.S., B.R., and B.F.S. All authors have read and agreed to the published version of the manuscript.

**Funding:** This research was funded by the European Union's Horizon 2020 research and innovation program under the REMAP project, grant number 769288.

**Data Availability Statement:** The maintenance data used in this study is available at https://doi.org/10.4121/uuid:1630e6fd-9574-46e8-899e-83037c17bcef (accessed on 28 December 2020).

**Conflicts of Interest:** The authors declare no conflict of interest.

## Nomenclature

The following nomenclature is used in this manuscript:

| | |
|---|---|
| $N$ | total number of aircraft |
| $i$ | aircraft indicator |
| $j$ | step indicator |
| $J$ | final step |
| $t_0$ | initial day |
| $T$ | final day |
| $k$ | check type |
| $h$ | hangar indicator |
| $I_{i,k}^{DY}$ | interval of type $k$ check of aircraft $i$ in DY |
| $I_{i,k}^{FH}$ | interval of type $k$ check of aircraft $i$ in FH |
| $I_{i,k}^{FC}$ | interval of type $k$ check of aircraft $i$ in FC |
| $\theta_{i,k}^{DY}$ | maximum tolerance of type $k$ check of aircraft $i$ in DY |
| $\theta_{i,k}^{FH}$ | maximum tolerance of type $k$ check of aircraft $i$ in FH |
| $\theta_{i,k}^{FC}$ | maximum tolerance of type $k$ check of aircraft $i$ in FC |
| $L_i^k$ | next type $k$ check duration (in DY) of aircraft $i$ |
| $w_k$ | minimum number of DY between the start of two type $k$ checks |
| $fh_i$ | average daily FH usage of aircraft $i$ |
| $fc_i$ | average daily FC usage of aircraft $i$ |
| $DY_{i,t}^k$ | cumulative DY of aircraft $i$ on day $t$ since its last type $k$ check |
| $FH_{i,t}^k$ | cumulative FH of aircraft $i$ on day $t$ since its last type $k$ check |
| $FC_{i,t}^k$ | cumulative FC of aircraft $i$ on day $t$ since its last type $k$ check |
| $\sigma_{i,t}^{DY}$ | tolerance used in the last type $k$ check of aircraft $i$ in DY |
| $\sigma_{i,t}^{FH}$ | tolerance used in the last type $k$ check of aircraft $i$ in FH |
| $\sigma_{i,t}^{FC}$ | tolerance used in the last type $k$ check of aircraft $i$ in FC |
| $m_{h,t}^k$ | binary variable to indicate if a type $k$ check can be performed in hangar $h$ on day $t$ |
| $M_t^k$ | hangar capacity for type $k$ check on day $t$ |
| $D_i^k$ | next type $k$ check due date of aircraft $i$ |
| $u_i^k$ | tolerance limit date for the next type $k$ check of aircraft $i$ |
| $y_i^k$ | end day of the last type $k$ check of aircraft $i$ |
| $z_i^k$ | scheduled day of the next type $k$ check of aircraft $i$ |
| $g_{i,t}^k$ | binary variable to indicate if aircraft $i$ is grounded on day $t$ performing a type $k$ check |
| $G_{i,t}^k$ | binary variable to indicate if there is a type $k$ check starting on day $t$ for aircraft $i$ |
| $G_t^k$ | binary variable to indicate if there is enough space between type $k$ checks when a type $k$ check is scheduled on day $t$ |
| $H_{i,t}^k$ | binary variable to indicate if there is enough hangar capacity to schedule a type $k$ check of aircraft $i$ on day $t$ |
| $P$ | penalty for using the tolerance when scheduling a check |
| $\delta_i$ | FH lost in the last scheduled check of aircraft $i$ |
| $x_j$ | decision variable |
| $s_j$ | state variable |
| $\pi$ | scheduling policy |
| $C_j(s_j, x_j)$ | cost of choosing action $x_j$ on state $s_j$ |

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
