# Peer review of "Aircraft Maintenance Check Scheduling Using Reinforcement Learning"

_aerospace, doi:10.3390/aerospace8040113_

Round 1

Reviewer 1 Report

The paper proposed a RL solution to optimise the long-term scheduling of A-check and C-checks for an aircraft fleet. The background and related literature are well introduced. The research design and methodology are clear. The experiment settings are reasonable. The results demonstrated the proposed model is able to achieve the design goals. The comparison study shows the good performance of the proposed model. The article is well written. I really enjoy reading this article. 

Reviewer 2 Report

This work shows in a very clear way how the challenging problem of Aircraft maintenance check scheduling can be addressed successfully by Reinforcement Learning. 

The main contribution is to develop a Deep Q-learning algorithm to solve the challenging check scheduling problem and establish the fact that after training the time to obtain a new solution is in the order of seconds.  The accuracy and stability of the solution are verified adequately and in a novel way. 

The work paves the way to solve challenging problems in maintenance where historical data can be exploited and motivates ideas to generate additional databases during maintenance. 

Could this Deep Q-Learning formulation of data-driven problems be applied to the C-check where the aircraft structure is inspected?  Here the data could stem from vibration-based monitoring of the aircraft structure and the machinery. 

In a broad sense, the problem is a Dynamics Problem and a solution is computed in a machine learning framework formulation which is quite different from the classical model-based dynamical problem. Thus is the solution presented in Figures 3 and 4 unique? Could we have a bifurcation and have multiple nearby solutions? 

Reviewer 3 Report

This paper proposed a reinforcement learning algorithm to oprimized aricraft fleet scheduling.
The proposed approach is evaluated very well in mathematics and real scenario.
And also the results clearly presented.

Reviewer 4 Report

The paper is well structured. It explains the background and the purpose of the paper succinctly. Authors have performed comparison with DP and proved that RL method is superior in scheduling the aircraft maintenance.